# Joint rotational invariance and adversarial training of a dual-stream Transformer yields state of the art Brain-Score for Area V4

## Abstract

Modern high-scoring models of vision in the brain score competition do not stem from Vision Transformers. However, in this paper, we provide evidence against the unexpected trend of Vision Transformers (ViT) being not perceptually aligned with human visual representations by showing how a dual-stream Transformer, a CrossViT *a la* Chen et al. (2021), under a joint rotationally-invariant and adversarial optimization procedure yields 2nd place in the aggregate Brain-Score 2022 competition (Schrimpf et al., 2020b) averaged across all visual categories, and at the time of the competition held 1st place for the highest explainable variance of area V4. In addition, our current Transformer-based model also achieves greater explainable variance for areas V4, IT and Behavior than a biologically-inspired CNN (ResNet50) that integrates a frontal V1-like computation module (Dapello et al., 2020). To assess the contribution of the optimization scheme with respect to the CrossViT architecture, we perform several additional experiments on differently optimized CrossViT's regarding adversarial robustness, common corruption benchmarks, mid-ventral stimuli interpretation and feature inversion. Against our initial expectations, our family of results provides tentative support for an *"All roads lead to Rome"* argument enforced via a joint optimization rule even for non biologically-motivated models of vision such as Vision Transformers.

## 1   Optimizing a CrossViT for the Brain-Score Competition

In this short paper, we try to solve an interesting question that was one of the motivations of this work: *"Are Vision Transformers good models of the human ventral stream?"* Our approach to answering this question will rely on using the Brain-Score platform (Schrimpf et al., 2020a) and participating in their first yearly competition with a Transformer-based model. This platform quantifies the similarity via bounded [0,1] scores of responses between a computer model and a set of non-human primates. Here the ground truth is collected via neurophysiological recordings and/or behavioral outputs when primates are performing psychophysical tasks, and the scores are computed by some derivation of Representational Similarity Analysis (Kriegeskorte et al., 2008) when pitted against artificial neural network activations of modern computer vision models.

We discuss an interesting finding, where amidst the constant debate of the biological plausibility of Vision Transformers – which have been deemed less biologically plausible than convolutional neural networks (as discussed in: URL_1 URL_2, though also see Conwell et al. (2021)) –, we find that when these Transformers are optimized under certain conditions, they may achieve high explainable variance with regards to many areas in primate vision, and surprisingly the highest score to date at the time of the competition for explainable variance in area V4, that still remains a mystery in visual

Submitted to 4th Workshop on Shared Visual Representations in Human and Machine Visual Intelligence (SVRHM) at NeurIPS 2022. Do not distribute.

| Rank | Model ID # | Description | Brain-Score | | | | | | $\rho$-Hierarchy |
|---|---|---|---|---|---|---|---|---|---|
| | | | Avg | V1 | V2 | V4 | IT | Behavior | |
| 1 | 1033 | Bag of Tricks (Riedel, 2022) [New SOTA] | **0.515** | **0.568** | **0.360** | 0.481 | 0.514 | **0.652** | -0.2 |
| 2 | 991 | CrossViT-18† (Adv + Rot) [Ours] | 0.488 | 0.493 | 0.342 | **0.514** | **0.531** | 0.562 | **+0.8** |
| 3 | 1044 | Gated Recurrence (Azeglio et al., 2022) | 0.463 | 0.509 | 0.303 | 0.482 | 0.467 | 0.554 | -0.4 |
| 4 | 896 | N/A | 0.456 | 0.538 | 0.336 | 0.485 | 0.459 | 0.461 | -0.4 |
| 5 | 1031 | N/A | 0.453 | 0.539 | 0.332 | 0.475 | 0.510 | 0.410 | -0.2 |

Table 1: Ranking of all entries in the Brain-Score 2022 competition as of February 28th, 2022. Scores in **blue** indicate **world record** (highest of all models at the time of the competition), while scores in **bold** display the highest scores of **competing entries**. Column $\rho$-Hierarchy indicates the Spearman rank correlation between per-Area Brain-Score and Depth of Visual Area (V1 $\rightarrow$ IT).

neuroscience (see Pasupathy et al. (2020) for a review). Our final model and highest scoring model was based on several insights:

**Adversarial-Training**: Work by Santurkar et al. (2019); Engstrom et al. (2019b); Dapello et al. (2020), has shown that convolutional neural networks trained adversarially[1] yield human perceptually-aligned distortions when attacked. This is an interesting finding, that perhaps extends to vision transformers, but has never been qualitatively tested before though recent works – including this one (See Figure 2) – have started to investigate in this direction (Tuli et al., 2021; Caro et al., 2020). Thus we projected that once we picked a specific vision transformer architecture, we would train it adversarially.

**Multi-Resolution**: Pyramid approaches (Burt & Adelson, 1987; Simoncelli & Freeman, 1995; Heeger & Bergen, 1995) have been shown to correlate highly with good models of Brain-Scores (Marques et al., 2021). We devised that our Transformer had to incorporate this type of processing either implicitly or explicitly in its architecture.

**Rotation Invariance**: Object identification is generally rotationally invariant (depending on the category; *e.g.* not the case for faces (Kanwisher et al., 1998)). So we implicitly trained our model to take in different rotated object samples via hard rotation-based data augmentation. This procedure is different from pioneering work of Ecker et al. (2019) which explicitly added rotation equivariance to a convolutional neural network.

**Localized texture-based computation**: Despite the emergence of a *global* texture-bias in object recognition when training Deep Neural Networks (Geirhos et al., 2019) – object recognition is a compositional process (Brendel & Bethge, 2019; Deza et al., 2020). Recently, works in neuroscience have also suggested that *local* texture computation is perhaps pivotal for object recognition to either create an ideal basis set from which to represent objects (Long et al., 2018; Jagadeesh & Gardner, 2022) and/or encode robust representations (Harrington & Deza, 2022).

After searching for several models in the computer vision literature that resemble a Transformer model that ticks all the boxes above, we opted for a CrossViT-18† (that includes multi-resolution + local texture-based computation) that was trained with rotation-based augmentations and also adversarial training (See Appendix A.3 for exact training details, our *best* model also used $p = 0.25$ grayscale augmentation, though this contribution to model Brain-Score is minimal).

Table 2: Selected Layers of CrossViT-18†

| Benchmark | Layer |
|---|---|
| V1,V2,V4 | blocks.1.blocks.1.0.norm1 |
| IT | blocks.1.blocks.1.4.norm2 |
| Behavior | blocks.2.revert_projs.1.2 |

**Results:** Our best performing model #991 achieved 2nd place in the overall Brain-Score 2022 competition (Schrimpf et al., 2020b)) as shown in Table 1. At the time of submission, it holds the first place for the highest explainable variance of area V4 and the second highest score in the IT area. Our model also currently ranks 6th across all Brain-Score submitted models as shown on the main brain-score website (including those outside the competition and since the start of the platform's conception, totaling 216). A general schematic of how Brain-Scores are calculated can be seen in Figure 1.

---

[1]Adversarial training is the process in which an image in the training distribution of a network is perturbed adversarially (*e.g.* via PGD); the perturbed image is re-labeled to its original non-perturbed class, and the network is optimized via Empirical Risk Minimization (Madry et al., 2018).

| | | ImageNet (↑) | Brain-Score (↑) | | | | | |
|---|---|---|---|---|---|---|---|---|
| Model ID # | Description | Validation Accuracy (%) | Avg | V1 | V2 | V4 | IT | Behavior |
| N/A | Pixels (Baseline) | N/A | 0.053 | 0.158 | 0.003 | 0.048 | 0.035 | 0.020 |
| N/A | AlexNet (Baseline) | 63.3 | 0.424 | 0.508 | 0.353 | 0.443 | 0.447 | 0.370 |
| N/A | VOneResNet50-robust (SOTA) | 71.7 | **0.492** | **0.531** | **0.391** | 0.471 | 0.522 | 0.545 |
| 991 | CrossViT-18† (Adv + Rot) | 73.53 | 0.488 | 0.493 | 0.342 | **0.514** | **0.531** | **0.562** |
| 1084 | CrossViT-18† (Adv) | 64.60 | 0.462 | 0.497 | 0.343 | 0.508 | 0.519 | 0.441 |
| 1095 | CrossViT-18† (Rot) | 79.22 | 0.458 | 0.458 | 0.288 | 0.495 | 0.503 | 0.547 |
| 1057 | CrossViT-18† | **83.05** | 0.442 | 0.473 | 0.274 | 0.478 | 0.484 | 0.500 |

Table 3: A list of different models submitted to the Brain-Score 2022 competition. Scores in **bold** indicate the highest performing model per column. Scores in **blue** indicate **world record** (highest of all models at the time of the competition). All CrossViT-18† entries in the table are ours.

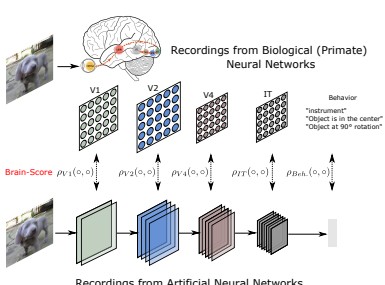

Figure 1: A schematic of how brain-score is calculated as similarity metrics obtained from neural responses and model activations.

Additionally, in comparison with the biologically-inspired model (VOneResNet50+ Adv. training), our model achieves greater scores in the IT, V4 and Behavioral benchmarks. Critically we notice that our best-performing model (#991) has a *positive* $\rho$-Hierarchy coefficient[2] compared to the new state of the art model (#1033) and other remaining entries, where this coefficient is negative. This was an unexpected result that we found as most biologically-driven models obtain higher Brain-Scores at the initial stages of the visual hierarchy (V1) (Dapello et al., 2020), and these scores decrease as a function of hierarchy with generally worse Brain-Scores in the final stages (*e.g.* IT).

We also investigated the differential effects of rotation invariance and adversarial training used on top of a pretrained CrossViT-18† as shown in Table 3. We observed that each step independently helps to improve the overall Brain-Score, quite ironically at the expense of ImageNet Validation accuracy (Zhang et al., 2019). Interestingly, when both methods are combined (Adversarial training and rotation invariance), the model outperforms the baseline behavioral score by a large margin (+0.062), the IT score by (+0.047), the V4 score by (+0.036), the V2 score by (+0.068), and the V1 score by (+0.020). Finally, our best model also retains a great standard accuracy at ImageNet from its pretrained version albeit a 10% drop, yet the performance on ImageNet Validation Accuracy of our model (73.53%) is still greater than a more biologically principled model such as the adversarially trained VOneResNet-50 (71.7%) (Dapello et al., 2020).

## 2 Assessment of CrossViT-18†-based models

As we have seen that the *optimization* procedure heavily influences the brain-score of each CrossViT-18† model, and thus its alignment to human vision (at a coarse level accepting the premise of the Brain-Score competition). We will now explore how different variations of such CrossViT's change as a function of their training procedure, and thus their learned representations via a suite of experiments that are more classical in computer vision. Additional experiments with CrossViT-18†-based models can be seen at Appendix B.

### 2.1 Adversarial Attacks

One of our most interesting qualitative results is that the *direction* of the adversarial attack made on our highest performing model resembles a distortion class that seems to fool a human observer too (Figure 2). Alas, while the adversarial attack can be conceived as a type of *eigendistortion* as in Berardino et al. (2017) we *find* that the Brain-Score optimized Transformer models are more

---

[2]$\rho$-Hierarchy coefficient: We define this as the Spearman rank correlation between the Brain-Scores of areas [V1,V2,V4,IT] with hierarchy: [1,2,3,4]

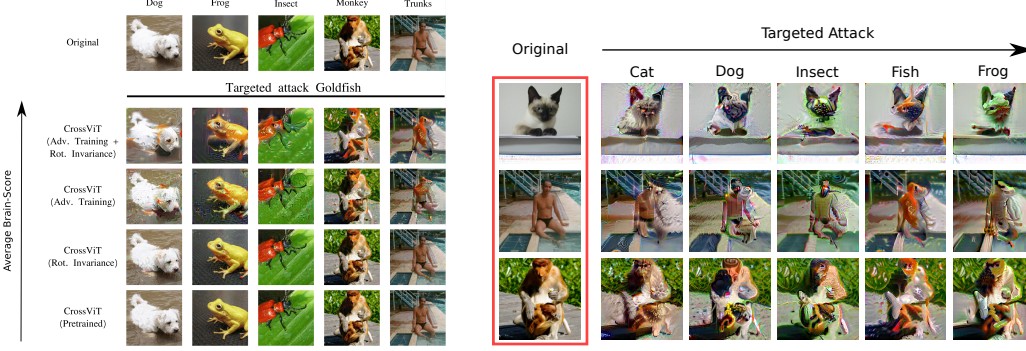

(a) A qualitative demonstration of the human-machine perceptual alignment of the CrossViT-18† via the effects of adversarial perturbations. As the average Brain-Score increases in our system, the distortions seem to fool a human as well.

(b) An extended demonstration of our winning model (CrossViT-18† [Adv. Training + Rot. invariance]) where a targeted attack is done for 3 images and the resulting stimuli is perceptually aligned with a human judgment of the fooled class.

Figure 2: Exploring Human-Machine Perceptual Alignment via Adversarial Attacks.

perceptually aligned to human observers when judging distorted stimuli. Similar results were previously found by Santurkar et al. (2019) with ResNets, though there has not been any rigorous & unlimited time verification of this phenomena in humans similar to the work of Elsayed et al. (2018).

## 2.2 Feature Inversion

The last assessment we provided was inspired by feature inversion models that are a window to the representational soul of each model (Mahendran & Vedaldi, 2015). Oftentimes, models that are aligned with human visual perception in terms of their inductive biases and priors will show renderings that are very similar to the original image even when initialized from a noise image (Feather et al., 2019). We use the list of stimuli from Harrington & Deza (2022) to compare how several of these stimuli look like when they are rendered from the penultimate layer of a pretrained and our winning entry CrossViT-based model. A collection of synthesized images can be seen in Figure 3.

Even when these images are rendered starting from different noise images, Transformer-based models are remarkably good at recovering the structure of these images. This hints at a coherence with the results of Tuli et al. (2021) who have argued that Transformer-based models have a stronger shape bias than most CNN's (Geirhos et al., 2019). We think this is due to their initial patch-embedding stage that preserves the visual organization of the image, though further investigation is necessary to validate this conjecture.

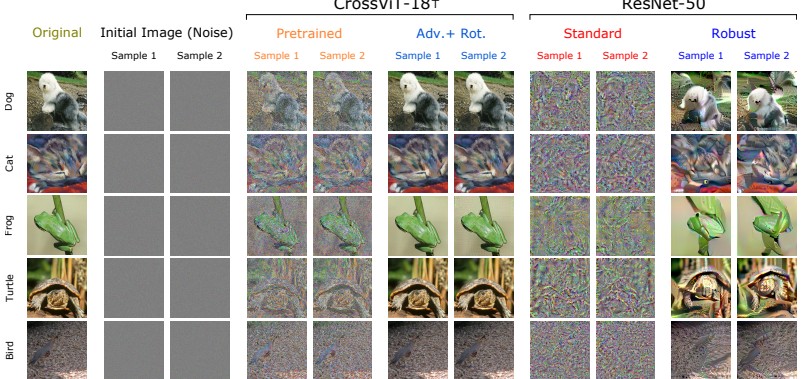

Figure 3: A summary of Feature Inversion models when applied on two different randomly samples noise images from a subset of the stimuli used in Harrington & Deza (2022). Standard and Pretrained models poorly invert the original stimuli leaving high spatial frequency artifacts. Adversarial training improves image inversion models, and this is even more evident for Transformer models.

## 3 Discussion

A question from this work that motivated the writing of this paper beyond the achievement of a high score in the Brain-Score competition is: How does a CrossViT-18† perform so well at explaining variance in primate area V4 without many iterations of hyper-parameter engineering? In this paper, we have only scratched the surface of this question, but some clues have emerged.

One possibility is that the cross-attention mechanism of the CrossViT-18† is a proxy for Gramian-like operations that encode local texture computation (vs global *a la* Geirhos et al. (2019)) which have been shown to be pivotal for object representation in humans (Long et al., 2018; Jagadeesh & Gardner, 2022; Harrington & Deza, 2022). This initial conjecture is corroborated by our image inversion effects (Section 2.2) where we find that CrossViT's preserves the structure stronger than Residual Networks (ResNets), while vanilla ViT's shows strong grid-like artifacts (See Figures 12, 13 in the supplementary material).

Equally relevant throughout this paper has been the critical finding of the role of the optimization procedure and the influence it has on achieving high Brain-Scores – even for non-biologically plausible architectures (Riedel, 2022). Indeed, the simple combination of adding rotation invariance as an implicit inductive bias through data-augmentation, and adding "worst-case scenario" (adversarial) images in the training regime seems to create a perceptually-aligned representation for neural networks (Santurkar et al., 2019).

On the other hand, the contributions to visual neuroscience from this paper are non-obvious. Traditionally, work in vision science has started from investigating phenomena in biological systems via psychophysical experiments and/or neural recordings of highly controlled stimuli in animals, to later verify their use or emergence when engineered in artificial perceptual systems. We are now in a situation where we have "by accident" stumbled upon a perceptual system that can successfully model (with half the full explained variance) visual processing in human area V4 – a region of which its functional goal still remains a mystery to neuroscientists (Vacher et al., 2020; Bashivan et al., 2019) –, giving us the chance to reverse engineer and dissect the contributions of the optimization procedure to a fixed architecture. We have done our best to pin-point a causal root to this phenomena, but we can only make an educated guess that a system with a cross-attention mechanism can *even under regular training* achieve high V4 Brain-Scores, and these are maximized when optimized with our joint adversarial training and rotation invariance procedure.

Ultimately, does this mean that Vision Transformers are good models of the Human Ventral Stream? We think that an answer to this question is a response to the nursery rhyme: *"It looks like a duck, and walks like a duck, but it's not a duck!"* One may be tempted to affirm that it is a duck if we are only to examine the family of in-distribution images from ImageNet at inference; but when out of distribution stimuli are shown to both machine and human perceptual systems we will have a chance to accurately assess their degree of perceptual similarity[3]. We can tentatively expand this argument further by studying adversarial images for both perceptual systems (See also Figure 4). Future images used in the Brain-Score competition that will better assess human-machine representational similarity should use these adversarial-like images to test if the family of mistakes that machines make are similar in nature than to the ones made by humans (See For example Golan et al. (2020)). If that is to be the case, then we are one step closer to building machines that can *see* like humans.

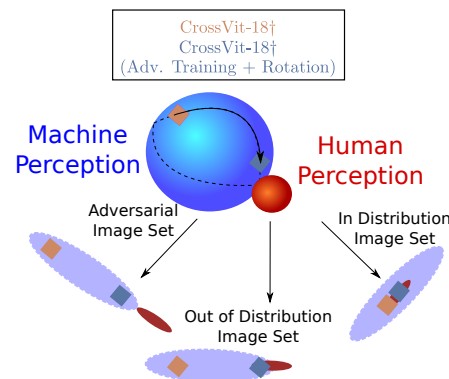

Figure 4: A cartoon inspired by Feather et al. (2019, 2021) depicting how our model changes its perceptual similarity depending on its optimization procedure. The arrows outside the spheres represent projections of such perceptual spaces that are observable by the images we show each system. While it may look like our winning model is "nearly human" it has still a long way to go, as the adversarial conditions have never been physiologically tested.

---

[3]Consider for example, that some stimuli used in Brain-Score are a basis set of Gabor filters, which are never encountered in nature

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

# A    Experimental Setup

## A.1    Dataset

We used the ImageNet 1k (Deng et al., 2009) dataset for training. ImageNet1K contains 1,000 classes and the number of training and validation images are 1.28 million and 50,000, respectively. We validate the effectiveness of our models in the different datasets proposed in the Brain-Score (Schrimpf et al., 2020a) competition.

## A.2    Custom Scheduler

The proposed learning rate scheduler is based on Jeddi et al. (2020) and is formulated as $LR = 0.00012 \times e - 0.0004$ for $e = 1$ and $LR = \frac{0.00002}{2^{e-2}}$ for $1 < e <= 6$. As shown in Figure 5, we start with a small learning rate and then it is smoothly increased for one epoch. We empirically found that fine-tuning the transformer for more than 1 epoch resulted in an under-fitting behavior of the adversarial robustness. After this first epoch, the learning rate is reduced very fast so that model performance converges to a steady state, without having too much time to overfit on the training data.

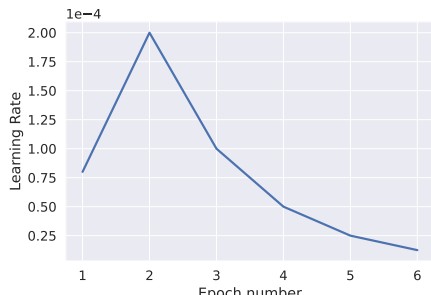

Figure 5: Custom scheduler used for training the Vision Transformer.

## A.3    Training Setup

We used a pretrained CrossViT-18† (Chen et al., 2021) downloaded from the timm library that is adversarially trained via a fast gradient sign method (FGSM) attack and random initialization (Wong et al., 2020). We opted for this strategy, known as "Fast Adversarial Training" as it allows a faster iteration in comparison with other common approaches (*e.g.* adversarial training with the PGD attack). In particular, all experiments used $\epsilon = 2/255$ and step size $\alpha = 1.25\epsilon$ as proposed originally in (Wong et al., 2020). However, in contrast to the previous method, we follow a 5 epoch fine-tuning approach with a custom learning rate scheduler in order to avoid underfitting. We optimize our networks with Adaptive Moment Estimation (Adam *a la* Kingma & Ba (2014)) and employed mixed precision for faster training. All input images were pre-

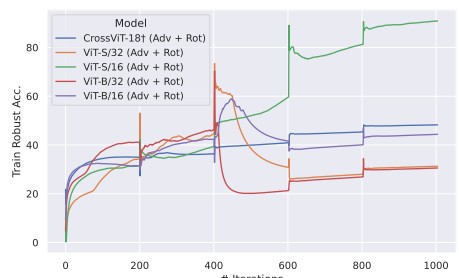

Figure 6: Training robust acc. of each Vision Transformer model (Adv + Rot). We clearly observed that ViT-S/16 has over-fitted during training.

processed with resizing to $256 \times 256$ followed by standard random cropping and horizontal mirroring. In the case of our best performing model (#991), we additionally incorporated a random grayscale transformation ($p = 0.25$) and a set of hard rotation transformations of (0°, 90°, 180°, 270°) – implicitly aiding for rotational invariance – due to the characteristics of images appearing in the behavioral benchmark of Rajalingham et al. (2018). All our experiments were ran locally on a GPU-Tesla V-100. Each adversarial training of a vision transformer took around 48 hours.

Optionally include extra information (complete proofs, additional experiments and plots) in the appendix. This section will often be part of the supplemental material.

# B  Additional Assessment of CrossViT-18†-based models

## B.1  Robustness against adversarial attacks

We also applied PGD attacks on our winning entry model (Adversarial Training + Rot. Invariance) on range $\epsilon \in \{1/255, 2/255, 4/255, 6/255, 8/255, 10/255\}$ and step-size = $\frac{2.5}{\#PGD_{iterations}}$ as in the robustness Python library (Engstrom et al., 2019a) , in addition to three other controls: Adv. Training, Rotational Invariance, and a pretrained CrossViT, to evaluate how their adversarial robustness would change as a function of this particular distortion class. When doing this evaluation we observe in Figure 7 that Adversarially trained models are more robust to PGD attacks (three-step size flavors: 1 (FGSM), 10 & 20). One may be tempted to say that this is "expected" as the adversarially trained networks would be more robust, but the type of adversarial attack on which they are trained is different (FGSM as part of FAT (Wong et al., 2020) during training; and PGD at testing). Even if FGSM can be interpreted as a 1 step PGD attack, it is not obvious that this type of generalization would occur. In fact, it is of particular interest that the Adversarially trained CrossViT-18† with "fast adversarial training" (FAT) shows greater robustness to PGD 1 step attacks when the epsilon value used at testing time is very close to the values used at training (See Figure 7a). Naturally, for PGD-based attacks where the step size is greater (10 and 20; Figs. 7b,7c), our winning entry model achieves greater robustness against all other trained CrossViT's independent of the $\epsilon$ values.

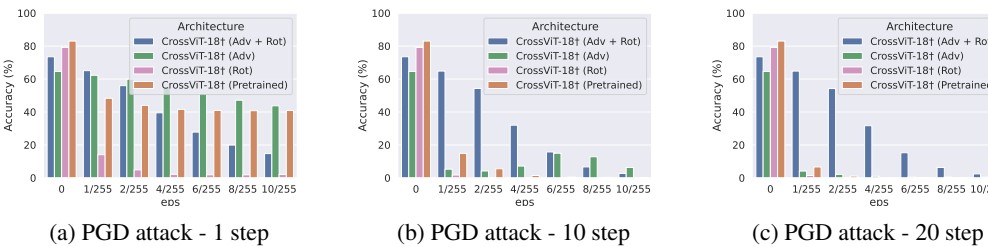

(a) PGD attack - 1 step      (b) PGD attack - 10 step      (c) PGD attack - 20 step

Figure 7: A suite of multiple steps [1,10,20] PGD-based adversarial attacks on clones of CrossViT-18† models that were optimized differently. Here we see that our winning entry (Adversarial training + Rotation Invariance) shows greater robustness (adversarial accuracy) than all other models as the number of steps of PGD-based attacks increases only for big step sizes of 10 & 20.

## B.2  Mid-Ventral Stimuli Interpretation

In addition to the previous experiments, we wondered how well the two models: CrossViT-18† (PreTrained) and CrossViT-18† (Adv. Training + Rot. Invariance) could linearly separate a small subset of 2-class stimuli across their visual hierarchy. For this experiment, we used both the original and texform stimuli (100 images per class) from Harrington & Deza (2022), where the texform stimuli can be used to test the mechanisms of human peripheral computation (Rosenholtz et al., 2012; Freeman & Simoncelli, 2011) or mid-ventral human computation (Long et al., 2018; Jagadeesh & Gardner, 2022). Roughly speaking these texforms are very similar to their original counter-part, where they match in global structure (*i.e.* form), but are locally distorted through a texture-matching operation (*i.e. texture*) as seen in Figure 8 (Inset 0.). In this analysis, we will use a t-SNE projection with a fixed random seed across both models and stimuli to evaluate the qualitative similarity/differences of their 2D clustering patterns.

Here we are interested in exposing our models to this distortion class because recent work has used these types of stimuli to show that human peripheral computation may act as a biological proxy for an adversarially robust processing system (Harrington & Deza, 2022), and that humans may in-fact use strong texture-like cues to perform object recognition (in IT) without the specific need for a strong structural cue (Jagadeesh & Gardner, 2022).

We find that Pretrained CrossViT-18† models have trouble in early visual cortex read-out sections to cluster both classes. In fact, several images are considered "visual outliers" for both original and texform images. These differences are slowly resolved only for the original images as we go higher in depth in the Transformer model until we get to the Behavior read-out layer. This is not the case for the texforms, where the PreTrained CrossViT-18† can not tease apart the primate and insect classes

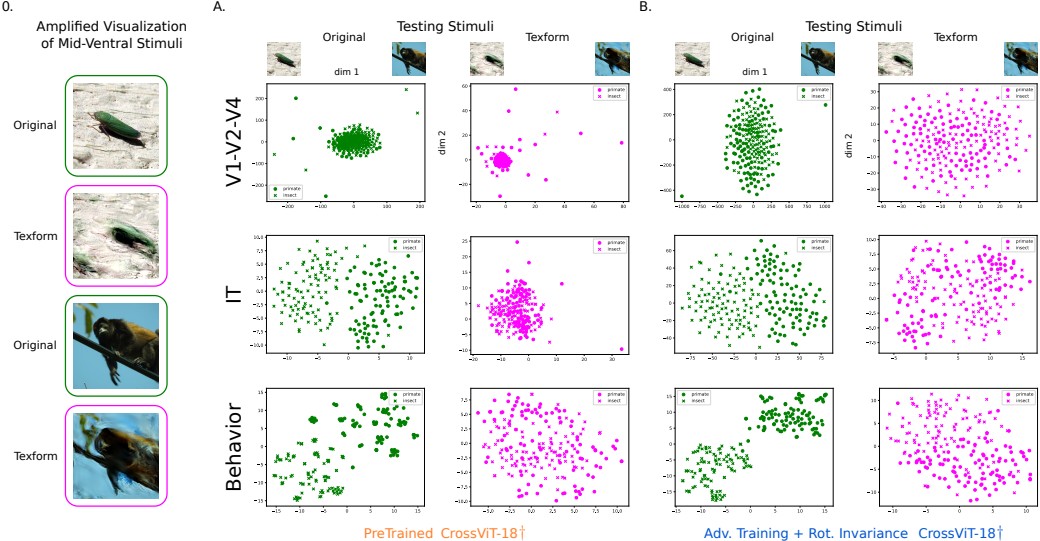

Figure 8: A comparison of how two CrossViT-18† models manage to classify original and texform stimuli. In (0.) we see a magnification of a texform, and in (A.,B.) we see how our winning Model Adv. + Rot. manages to create tighter vicinities across the visual stimuli, and ultimately – at the behavioral level – can separate both original and texform stimuli, while pretrained transformers seem to struggle with texform linear separability at the behavioral stage.

at such simulated behavioral stage. This story was to our surprise very different and more coherent with human visual processing for the Adv + Rot CrossViT-18† where outliers no longer exist – as there are none in the small dataset –, and the degree of linear separability for the original and texform stimuli increases to near perfect separation for both stimuli at the behavioral stage.

## B.3 Common Corruption Benchmarks

We also looked into how adversarial training would affect the performance of the different sets of neural networks to common corruptions that are *not* adversarial. To do this, we ran our models and benchmarked them to the ImageNet-C dataset (Hendrycks & Dietterich, 2019).

One would have expected Brain-Aligned models like our adversarially-trained + rotationally invariant CrossViT to also present strong robustness to common corruptions. To our surprise, this was not the case as seen in Table 5. This is a puzzling result, though there have been several bodies of work suggesting that adversarial robustness and common corruptions robustness are independent phenomena (Laugros et al., 2019), however, Kireev et al. (2021) have proved otherwise contingent on the $l_\infty$ radius [4] – but now see Li et al. (2022).

| Network | Clean Accuracy (↑) | mce (↓) | Gauss | Shot | Impulse | Defocus | Glass | Motion | Zoom | Snow | Frost | Fog | Bright | Contrast | Elastic | Pixel | JPEG |
|---|---|---|---|---|---|---|---|---|---|---|---|---|---|---|---|---|---|
| ResNet50-Augmix | 77.53 | 67.1 | 65.5 | 65.1 | 66.4 | 67.7 | 81 | 63.9 | 65.5 | 71.6 | 70.9 | 66.5 | 57.8 | 60.2 | 76.9 | 59.5 | 68.5 |
| CrossViT-18† (Adv + Rot) | 73.53 | 79.5 | 80.7 | 81.6 | 83.2 | 90.2 | 78.7 | 82.4 | 80 | 77.6 | 74 | 107.9 | 65 | 100.4 | 74.2 | 57.4 | 58.7 |
| CrossViT-18† (Adv) | 64.60 | 88.8 | 85 | 85.7 | 86.7 | 96.7 | 88 | 92.1 | 91.3 | 85.8 | 83.6 | 109.3 | 82.2 | 104.9 | 90 | 70.3 | 80.9 |
| CrossViT-18† (Rot) | 79.22 | 73.1 | 75.4 | 76.7 | 75 | 75.7 | 85.3 | 72.3 | 79.2 | 68.8 | 70.9 | 64.3 | 54.7 | 67.6 | 78.4 | 75.4 | 76.4 |
| CrossViT-18† | **83.05** | **51** | **46.1** | **48.8** | **46.4** | **61.2** | **72.6** | **54.4** | **65** | **44.9** | **42.1** | **37.2** | **41.5** | **37** | **67.2** | **46.8** | **54.2** |

Table 4: A table showing the comparison of mean corruption errors (mce)'s across CrossViT models contingent on their training regime. A ResNet50-Augmix is shown as a reference of a particularly strong model to common corruptions. Here lower scores are indicative of better robustness to the different distortion types of Hendrycks & Dietterich (2019).

---

[4]Also see Li et al. (2022) that shows that generally robust models (robust to adversarial + commmon corruptions) have a preference for low-spatial frequency statistics.

## B.4 ImageNet-R

We also looked into how adversarial training would affect the performance of generalization to various abstract visual renditions. To do this, we ran our models and benchmarked them on the ImageNet-Rendition (ImageNet-R) dataset (Hendrycks et al., 2021).

We observe that the accuracy on ImageNet-R decreases when the CrossViT is adversarially trained. However, when we combine the rotation invariance and adversarial training regimes, the accuracy on ImageNet-R becomes competitive with its pretrained version. In addition, we also appreciate that this combination does not affect the IID/OOD Gap with respect to the pretrained CrossViT.

| Network | ImageNet-200 (↑) | ImageNet-R (↑) | Gap (↓) |
|---------|------------------|----------------|---------|
| CrossViT-18† (Adv + Rot) | 90.75 | 41.14 | **49.61** |
| CrossViT-18† (Adv) | 85.52 | 35.73 | 49.79 |
| CrossViT-18† (Rot) | 93.89 | 37.35 | 56.54 |
| CrossViT-18† | 95.64 | **45.7** | 49.94 |

Table 5: A table showing the comparison of the accuracy on Imagenet-R dataset across CrossViT models contingent in their training regime.

## C   Comparison of CrossViT vs vanilla Transformer (ViT) Models

In this section, we investigated what is the role of the architecture in our results. Did we arrive at a high-scoring Brain-Score model by virtue of the general Transformer architecture, or was there something particular about the CrossViT (dual stream Transformer), that in tandem with our training pipeline allowed for a more ventral-stream like representation? We repeated our analysis and training procedures with a collection of vanilla Vision Transformers (ViT) where we manipulated the patch size and number of layers with the conventions of Dosovitskiy et al. (2021) as shown in Figure 9.

Here we see that the Brain-Score on V2, V4, superior processing IT, Behavior and Average *increase* independent of the type of Vision Transformer used for our suite of models (CrossViT-18†, and multiple ViT flavors) except for the particular case of ViT-S/16 due to over-fitting (See Figure 6) that heavily reflects on the behavior score. To our surprise, adversarial training in some cases helped V1 score and in some not, potentially due to an interaction with both patch size and transformer depth that has not fully been understood. In addition, to our knowledge, this is also the first time that it has been shown that adversarial training coupled with rotational invariance homogeneously increases brain-scores across Transformer-like architectures, as previous work has shown that classical CNNs (*i.e.* ResNets) increase Brain-Scores with adversarial training (Dapello et al., 2020). Additionally to the experiments on CrossViT-18†, we also evaluate the brain-scores on vanilla Vision transformers that can be seen in Table 6.

| | ImageNet(↑) | Brain-Score(↑) | | | | | |
|---|---|---|---|---|---|---|---|
| Description | Validation Acc. (%) | Avg | V1 | V2 | V4 | IT | Behavior |
| ViT-S/16 | 81.40 | 0.445 | 0.527 | 0.295 | 0.454 | 0.449 | 0.498 |
| ViT-S/32 | 75.99 | 0.415 | 0.531 | 0.271 | 0.422 | 0.423 | 0.426 |
| ViT-B/16 | 84.53 | 0.451 | 0.522 | 0.317 | 0.398 | 0.487 | 0.529 |
| ViT-B/32 | 80.72 | 0.440 | 0.553 | 0.311 | 0.413 | 0.418 | 0.505 |
| ViT-S/16 (Adv + Rot) | 50.44 | 0.443 | 0.506 | 0.332 | 0.470 | 0.496 | 0.409 |
| ViT-S/32 (Adv + Rot) | 55.20 | 0.457 | 0.512 | 0.347 | 0.433 | 0.485 | 0.508 |
| ViT-B/16 (Adv + Rot) | 67.25 | 0.486 | 0.536 | 0.332 | 0.470 | 0.496 | 0.598 |
| ViT-B/32 (Adv + Rot) | 53.01 | 0.457 | 0.524 | 0.357 | 0.417 | 0.472 | 0.515 |

Table 6: ImageNet accuracy, Brain-Scores of each brain area & Behavior benchmark evaluated on vanilla vision transformers

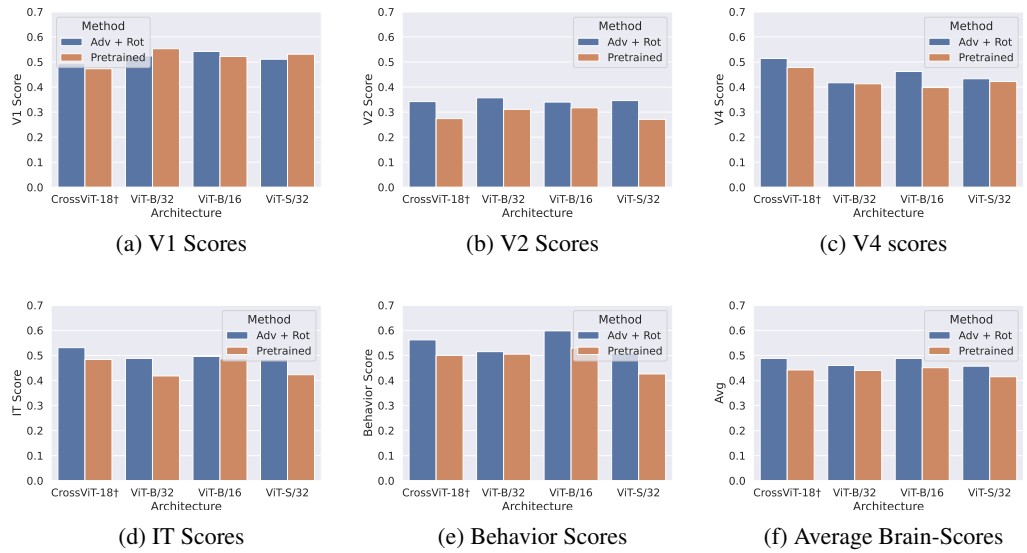

|  |  |  |
|:---:|:---:|:---:|
| (a) V1 Scores | (b) V2 Scores | (c) V4 scores |
| (d) IT Scores | (e) Behavior Scores | (f) Average Brain-Scores |

Figure 9: Similarity Brain-Score analysis on the different cortical areas of the ventral stream for vanilla transformers (ViT) and CrossViT. For nearly all Transformer variations, Adversarial Training with Joint Rotational Invariance increases per Area and Average Brain-Scores.

## D  Selection of the Best-BrainScore layers

Best performing layers on each vision transformer were selected by a brute-force approach. We evaluate each layer of the vision transformer models on each brain region and behavior dataset and select the layer that got the best score on the public benchmarks (in order to avoid overfitting) proportioned by Brain-Score organization. After this step, the "Adv + Rot" & pretrained versions of each transformer are submitted to the competition fixing best performing layers (See Table 7 ). We achieved our highest score at the time of our 4th submission, which was the lowest number of submissions in the competition (the winner of the competition performed nearly 60 submissions). All our results reflect the private scores obtained by each vision transformer model.

| Model | V1 | V2 | V4 | IT | Behavior |
|---|---|---|---|---|---|
| CrossViT-18† | blocks.1.blocks.1.0.norm1 | blocks.1.blocks.1.0.norm1 | blocks.1.blocks.1.0.norm1 | blocks.1.blocks.1.4.norm2 | blocks.2.revert_projs.1.2 |
| ViT-S/16 | blocks.1.mlp.act | blocks.3.attn.proj | blocks.3.norm2 | blocks.9.norm1 | pre_logits |
| ViT-S/16 | blocks.1.mlp.act | blocks.3.attn.proj | blocks.3.norm2 | blocks.9.norm1 | pre_logits |
| ViT-S/32 | blocks.1.mlp.act | blocks.10.norm1 | blocks.2.mlp.act | blocks.10.norm1 | pre_logits |
| ViT-B/16 | blocks.1.mlp.act | blocks.6.norm2 | blocks.2.mlp.act | blocks.8.norm1 | pre_logits |
| ViT-B/32 | blocks.1.mlp.act | blocks.6.norm2 | blocks.2.mlp.act | blocks.11.norm1 | pre_logits |

Table 7: Layers selected for each brain region on each vision transformer.

