# OpenReview forum: "Joint rotational invariance and adversarial training of a dual-stream Transformer yields state of the art Brain-Score for Area V4"
_NeurIPS.cc/2022/Workshop/SVRHM — SVRHM Poster_

### Official Review · Reviewer_sg7T · 2022-10-05
**Potential interesting, but quite hard to read**

**Rating:** 6
**Confidence:** 3

**Review:**

Overall the paper may have interesting results and insights, but it’s somehow hard to read and not very clear. Below are some pointers to the most obscure parts.

In the abstract, it’s quite unclear what an ‘all roads lead to Rome’ argument would be.

In general, it would be beneficial for the paper to define and introduce all the relevant concepts in advance (even with just one sentence): it may not be clear to all readers what adversarial training is, multi-resolution, etc.

The authors mention that one of their models (but it’s not clear how many they tested) has a positive p-Hierarchy, without though specifying what this implies or means in practice. This reviewer assumed this means that the model explains best higher regions than lower regions. And if so, why is this important and how can this be interpreted?

Section 2.1 adversarial attack: based on which results the authors concluded that ‘..seems to fool a human observer’? I couldn’t find any data in figure 2 to support this conclusion.

Something to consider discussing would concern the conclusion: a network is not necessarily a good model of the visual system just because it explains high variance, but it should be grounded on biologically sound inductive biases. For example: is really adversarial training biologically plausible?

I wouldn’t state that area V4 is a mystery to neuroscientists. Please refer to a few papers to unravel part of the mystery:
Anna W.Roe, LeonardoChelazzi, Charles E.Connor, Bevil R.Conway, IchiroFujita, Jack L.Gallant, HaidongLu, WimVanduffel "Toward a Unified Theory of Visual Area V4" Neuron, Volume 74, Issue 1, (2012), Pages 12-29
Pasupathy, A., Connor, C. "Population coding of shape in area V4." Nat Neurosci 5, 1332–1338 (2002). https://doi.org/10.1038/972
Pasupathy A, Popovkina DV, Kim T. Visual Functions of Primate Area V4. Annu Rev Vis Sci. 2020 Sep 15;6:363-385. doi: 10.1146/annurev-vision-030320-041306. Epub 2020

Very minor grammatical detail regarding the French ‘à la’: the ‘a’ has an accent.

---

### Official Review · Reviewer_L3M9 · 2022-10-14
**Vision transformer models are good models of ventral visual processing**

**Rating:** 7
**Confidence:** 3

**Review:**

### Summary
The authors sought to understand how well vision transformers, that appear not to be biologically plausible, can match visual responses in V1, V2, V4 and IT and behaviour. Using the brainscore benchmarking system, they found that a pre-trained multi-scale vision transformer that is then fine-tuned to be adversarially robust (using the fast-gradient sign method) along with rotation augmentations results a model that is highly competitive on the brainscore metric. They found that their model had very competitive V4 and IT response predictivity along with a very high behavioural score. Qualitatively, the authors found that targeted adversarial examples for CrossViT-18 (Adv+Rot) might also fool humans and that when using CrossViT-18 (Adv+Rot), images rendered from noise to match features from seed images were highly similar to the seed images themselves (i.e., images rendered using the cross-attention vision transformer were more similar qualitatively to the seed image than those rendered using a ResNet-50).

Overall, I thought the paper was clear and straightforward and I thank the authors for addressing this interesting question.

### Some questions (in no particular order)
1. Does CrossViT improve robustness to adversarial attacks or common corruptions relative to ViT? I am curious about the effects of multi-scale representations on robustness.
2. What do you make of the fact that a single group of artificial neurons at layer "blocks.1.blocks.1.0.norm1" all predict V1, V2, V4 responses the best across all transformer layers? Are the sets of artificial neurons that best predict V1, V2 and V4 responses disjoint? This is very much unlike results in, e.g., ResNets, where you have a nice hierarchy of model layers that best predict V1 and V4 (see, e.g., Zhuang et al., 2021). The clear hierarchy in ResNets basically says that computations in V4 are several linear-nonlinear computations on top of V1 responses.
3. Could you explain in more detail the $\rho$-hierarchy score? Why is correlating the actual brainscore of the model for each visual area with [1,2,3,4] a good metric (based on footnote 2). For example, if a model has brainscores of [0.1,0.2,0.3,0.4], it would receive a $\rho$-hierarchy score of 1, but the scores themselves are very low. I would have thought that one could compute a hierarchy score by correlating the model layer number that best predicts each visual area with [1,2,3,4]. I, however, may be misunderstanding the metric.
4. Given that the brain-score of ViT-B/16 (Adv+Rot) is pretty much the same as that of CrossViT-18 (Adv+Rot) (Table 6) (in fact, it performs extremely well on the behaviour benchmark), I am curious as to why you think that multi-scale processing in the model is a good desideratum for models of visual processing. Of course, the primate visual system has some form of multi-scale processing, but I wonder what behavioural advantages it confers, if any.
5. If you trained (fine-tuned) many different architectures with rotation augmentations, would you expect to see improvements in V4 neural predictivity across the board? It seems to be the case for ViTs and CrossViTs, but what about ResNets, VGGs, etc.?
6. What was the dimensionality of the features used in the feature-inversion analysis for both the CrossViT-18 and the ResNet-50 and what model layers were used in both models for feature inversion? My intuition is that if the dimensionality is small, more spatial information would have to be "discarded", resulting in a synthetic image that won't align well with the seed image. Is there any way to perform feature inversion while controlling for the number of features required in the optimization procedure so that the synthetic-image comparisons are more fair?

### Minor comment
- Figures 12 and 13 do not exist in the manuscript (they are referred to on line 141).

### References
- Zhuang, C., Yan, S., Nayebi, A., Schrimpf, M., Frank, M. C., DiCarlo, J. J., & Yamins, D. L. (2021). Unsupervised neural network models of the ventral visual stream. Proceedings of the National Academy of Sciences, 118(3), e2014196118.

---

### Official Review · Reviewer_VU2H · 2022-10-14
**It is a work that fits a bio-implausible machine learning architecture to vision. By switching from the kind of cost functions from which transformers arise, to those that can induce higher brain-score, the authors showed that transformers can be tuned to behave quite like IT and decently like V1. The result seems to suggest that while brains are constrained in many aspects, as the depth goes higher, eventually brains find a way to understand things the same as transformers do.**

**Rating:** 6
**Confidence:** 2

**Review:**

Quality: good
Clarity: very good
Originality: fitting a biologically implausible artificial neural network to a brain is an innovation in the sense that, while obvious, most people won't take this route. Adversarial training is recently known, rotational invariance was well known, training using those I regard as incremental innovation.
Significance: the discovery that at deeper level of brains, biological brains and a SOTA but biologically-implausible neural architecture reach the same conclusion is significant, at least to me. It seems to indicate that the recent improvements in Machine Learning for visual classification accuracy benchmark has likely produced artificial architectures that are superior to biological neural architectures, at least in those benchmarks. Thus, if the goal of studying biological neural architectures is to apply their good tricks in artificial systems, the community might be better served to shift our focus to properties such as power consumption as opposed to accuracy.